# Persistence to Medications for Benign Prostatic Hyperplasia/Benign Prostatic Obstruction-Associated Lower Urinary Tract Symptoms in the ASL TO4 Regione Piemonte (Italy)

**DOI:** 10.3390/healthcare10122567

**Published:** 2022-12-17

**Authors:** Lucrezia Greta Armando, Raffaella Baroetto Parisi, Elisa Remani, Mariangela Esiliato, Cristina Rolando, Valeria Vinciguerra, Abdoulaye Diarassouba, Clara Cena, Gianluca Miglio

**Affiliations:** 1Department of Drug Science and Technology, University of Turin, Via Pietro Giuria 9, 10125 Turin, Italy; 2Struttura Complessa Farmacia Territoriale ASL TO4, Regione Piemonte, Via Po 11, 10034 Chivasso, Italy; 3Competence Centre for Scientific Computing, University of Turin, Corso Svizzera 185, 10149 Turin, Italy

**Keywords:** α_1_-adrenoceptor antagonists, steroid 5α-reductase inhibitors, adherence, drug prescription networks

## Abstract

Background: Pharmacological treatment of benign prostatic hyperplasia (BPH)/benign prostatic obstruction (BPO)-associated lower urinary tract symptoms (LUTS) aims at improving patients’ quality of life by managing urinary symptoms and preventing complications and disease progression. However, continuous use of drugs to treat BPH/BPO-associated LUTS decreases over time. The aim of this retrospective observational study was to describe use of α1-adrenoceptor antagonists (ABs) and steroid 5α-reductase inhibitors (5ARIs) by adult (age ≥ 40 years) men in the ASL TO4, a Local Health Authority in the northern area of the city of Turin (Italy). Methods: Persistence measures were adopted as a robust, informative, and feasible way to understand medication-taking behavior and to assess patient compliance. Results: A total of 4309 men (median age 71 years) were enrolled. Monotherapy was the treatment option prescribed to the largest part of the study population. However, ≥two drugs were prescribed to a substantial proportion of men (23%). Men prescribed alfuzosin or dutasteride had significantly greater persistence, which decreased over time. Conclusions: Unmet needs and areas of intervention for healthcare systems aimed at improving the use of drugs for BHP/BPO-associated LUTS in the ASL TO4 Regione Piemonte were identified.

## 1. Introduction

Pathologically diagnosed benign prostatic hyperplasia (BPH) is characterized by the non-malignant proliferation of the epithelial and stromal components of the prostate gland. Such changes may be associated with benign prostatic obstruction (BPO), which could result in a number of lower urinary tract symptoms (LUTS). These involve storage (irritative) symptoms, voiding (obstructive) symptoms and/or post-micturition symptoms, which could have a considerable impact on patients’ quality of life (QoL), psychological well-being, as well as interfering with their activities of daily living [1,2].

Medical treatment for BPH/BPO has evolved over the last few decades, with a growing interest in the pharmacological management of LUTS over more invasive approaches. For example, current recommendations for the treatment of LUTS include the adoption of behavioral and dietary interventions and of pharmacological therapies when these interventions are inappropriate or unsuccessful [3,4]. In particular, pharmacological treatment aims at improving patients’ QoL by managing urinary symptoms and preventing complications and disease progression [5,6]. Medications belonging to five pharmacological classes are usually considered: α_1_-adrenoceptor antagonists (α_1_-blockers, ABs); steroid 5α-reductase inhibitors (5ARIs); muscarinic receptor antagonists; phosphodiesterase 5 inhibitors; β_3_-adrenoceptor antagonists and herbal drug preparations. These agents can be used either as monotherapy or in combination. For example, ABs and 5ARIs were demonstrated to significantly relieve symptoms and/or progression of BPH/BPO in a substantial number of randomized clinical trials [7,8,9,10,11,12], and they are widely used in the management of BPH/BPO-associated LUTS. Nevertheless, several needs of patients remain unmet [13,14]. In addition, as demonstrated by a number of studies conducted in daily practice, continuous use of medications to treat BPH/BPO-associated LUTS typically decreases over time [15,16,17,18,19,20,21,22]. Discontinuation of these medications may lead to complications, including acute urinary retention, urinary tract infection, stone formation, neurogenic bladder and renal insufficiency, as well as affecting their effects on disease progression [15,19,20]. Multiple barriers to long-term continuation of medications for BPH/BPO-associated LUTS could be considered, including clinical factors (i.e., presence of only one type of symptom, normal prostate-specific antigen level and/or younger age), pharmacological factors (i.e., a drug’s pharmacokinetic–pharmacodynamic profile and/or tolerability), and factors associated with management of patient therapy (i.e., use of multiple medications and/or dose adjustments). In addition, a wide variation among different geographical areas and countries has been reported with regard to drug prescription patterns for different pharmacological classes [23]. This variation can be attributed to the cost of medications and to different healthcare policies, which could limit access to some drugs, as well as social and cultural differences. These issues could impact on patient medication adherence to different degrees; therefore, changing these factors could yield important improvements in the quality of care.

The aim of this retrospective, observational study was to describe the use of drugs for the treatment of BPH/BPO-associated LUTS in adult men in the ASL TO4, a Local Health Authority (LHA) in the northern area of the city of Turin in the Piedmont Region (Italy). In particular, the ASL TO4 includes both urban and rural areas, and it meets the health needs of approximately 505,000 inhabitants, representing 12% of the overall regional population. In particular, persistence measures were adopted as a robust, informative and feasible way to quantify patient compliance [24].

## 2. Materials and Methods

### 2.1. Data Source

The source of data was the electronic health records of the ASL TO4. These databases contain the records of drugs dispensed by local pharmacies that were reimbursed by the Italian National Health System (NHS) to all citizens in the LHA’s area. They include the date of birth and gender of the patient, the dispensation date, the number of dispensed packages, the name of the drug and its active ingredient, the Anatomical Therapeutic Chemical (ATC) code and the number of Defined Daily Doses (DDDs) [25]. Drug dispensing data proved to be a feasible data source for observational retrospective studies [26].

This study was carried out in compliance with the General Data Protection Regulation (EU) 2016/679 and the data were treated fully in accordance with current privacy legislation. In particular, data were anonymized at source and authors had no access to patient identifiable data. Formal ethical approval by the ASL TO4 Ethics Committee is not required for retrospective observational studies.

### 2.2. Study Population and Drugs

Eligible subjects were men aged 40 years or older on 1 April 2018. In addition, as the medication is intended for the treatment of a chronic condition, we included in the analysis only men who had at least 2 distinct prescriptions for study drugs in the follow-up period (chronic medication) and excluded those with only 1 prescription (occasional medication). The date of filling the first prescription during the index period marked the index date. A longitudinal dataset of medication supply was created for each patient and the number of days of drug supply was calculated based on the DDD; each subject was followed up for 365 days. Men with either less than 1 year of database history prior to the index date or who received prescriptions for a study drug between 1 January and 31 March 2018 (wash-out period) were excluded. Study drugs were identified according to their ATC code, and they belonged to either the AB (G04CA) or the 5ARI (G04CB) pharmacological classes. These drugs were chosen as their prescription is reimbursed by the Italian NHS; therefore, their use can be studied accurately by analyzing administrative databases.

### 2.3. Statistical Analysis

Baseline demographics and characteristics were reported descriptively. Treatment persistence was estimated using drug survival analysis (Kaplan–Meier survival analysis), where an individual’s persistence is expressed as the time from the index date to the first discontinuation of a study drug, defined as a gap between treatment episodes of drug use for each subject. For this purpose, the duration of each dispensation was estimated by dividing the total amount of active substance contained in each package dispensed by the DDD. A continued treatment episode was considered if a new dispensation occurred within an appropriate grace period [27]. In particular, because a variety of different medicinal products with different prescription duration were available on the Italian market for most of the study drugs, the length of the grace period in drug survival analyses was defined as 1.5 times the specific prescription duration. Comparisons of treatment persistence in the study population were performed for ABs and 5ARIs.

All analyses were performed using the R statistical and programming language (version 4.2.0; https://cran.r-project.org/, accessed on 1 May 2022). In addition, several add-on packages (and their dependencies) were used: lubridate, ggplot2, ggpubr, tidyverse, doBy, stringi, stringr, data.table, dplyr, survival, survminer, and Rcpp.

## 3. Results

### 3.1. Baseline Characteristics

A total of 149,087 men aged 40 years or older on 1 April 2018 was initially considered. Among them, 4309 met the inclusion/exclusion criteria and were considered. General characteristics of the study population are summarized in Table 1. The median age was 71.0 years [interquartile range, IQR, 64.0 to 78.0]. The largest subset (35.6%) were men in the age group 70–79 years, while less than 5.0% (180 men) were in the age groups at the extremities (Table 1). A total of 140 (3.2%) died during their follow-up periods, but they were not excluded because death was considered as a possible cause of non-persistence.

A total of seven different drugs were found in the study dataset: five belonging to the AB class (alfuzosin, doxazosin, silodosin, tamsulosin and terazosin) and two to the 5ARI class (finasteride and dutasteride).

### 3.2. Persistence to Medication

Drug dispensing data were analyzed to understand the behavior of the population of the ASL TO4 toward drugs to treat BPH/BPO-associated LUTS; for this purpose, persistence to medication was studied first. Median time to discontinuation was significantly longer (*p* < 0.001, log-rank test) for alfuzosin (278 days [95% CI, 240 to 335]) than for other ABs (silodosin, 120 days [95% CI, 104 to 128]; tamsulosin, 101 days [95% CI, 94 to 119]; doxazosin, 81 days [95% CI, 61 to 120]; terazosin, 28 days [95% CI, 28 to 56], Figure 1A). In addition, the percentage of men still in treatment at 365 days was 25.5% (95% CI, 24.2 to 26.9) for the overall ABs, but it ranged from 43.8% (95% CI, 39.8 to 47.9) to 4.5% (95% CI, 0.2 to 8.9) for alfuzosin and doxazosin, respectively (Table 2). Persistence to ABs was significantly higher for older men (≥70 years of age) than for younger men (<70 years of age) (median time to discontinuation, 120 days [95% CI, 105 to 123] vs. 106 days [96 to 120], respectively, *p* = 0.028, log-rank test, Figure 1B).

Median time to discontinuation was significantly (*p* < 0.001, log-rank test) longer for dutasteride (272 days [95% CI, 240 to 305]) than for finasteride (97 days [95% CI, 82 to 119], Figure 2A). Furthermore, the percentage of persistent men at 365 days was 36.0% (95% CI, 33.6 to 38.5) for the overall 5ARIs, but it was higher for dutasteride alone (43.3% [95% CI, 40.3 to 46.3]) than for finasteride alone (17.3% [95% CI, 13.7 to 20.9], Table 2). No significant difference (*p* = 0.95, log-rank test) was found when older and younger men were compared (median time to discontinuation, 185 days [95% CI, 167 to 233] vs. 188 days [95% CI, 171 to 221], respectively, Figure 2B). Collectively, these results demonstrate that persistence depends on drug-related factors. In addition, age-related factors could play a role.

### 3.3. Prescription of Drugs for BPH/BPO-Associated LUTS

To better understand persistence results, prescription patterns were further analyzed. A large proportion of men (91.4%) in the study population were prescribed drugs belonging to a single pharmacological class. At least one AB was prescribed to 3273 (76.0%) men, at least one 5ARI to 1407 (32.7%) men, and at least one AB in combination with at least one 5ARI was prescribed to 471 (10.8%) men. Men prescribed 5ARIs were significantly (*p* < 0.001, Wilcoxon rank sum test) older than those prescribed ABs (Table 3). The percentage of men prescribed a single drug was 76.0% (Appendix A). Tamsulosin (*N* = 2057; 47.7%) and dutasteride (*N* = 1064; 24.7%) were the AB and the 5ARI prescribed to the largest subsets of men (Table 3 and Appendix A). 

Twenty out of twenty-one possible drug pairs prescribed to the same individual were detected in the study population through the analysis of drug prescription networks. Tamsulosin–silodosin was the drug pair most commonly prescribed (*N* = 355; 8.2%) within the AB class; for 5ARIs, dutasteride–finasteride was prescribed to 73 (1.7%) men. Doxazosin–finasteride was not detected in the study population. As shown in Figure 3A and Appendix A, drugs belonging to a single pharmacological class were prescribed to men in the age group 40–49 years, while the percentage of men prescribed drugs belonging to two classes increased from the age group 50–59 years (4.3%) to the age group 80–89 years (11.5%). The largest subset of men was prescribed a single drug across all the age groups; about one fifth were prescribed two different drugs, while less than 5.0% were prescribed ≥ three drugs (Figure 3B and Appendix A). Interestingly, the percentage of men prescribed ≥ two drugs was higher for the younger age groups (men < 70 years of age), and lower for the older age groups. These findings suggest that the prescription of drugs for the treatment of BPH/BPO-associated LUTS could be influenced by life transitions. To obtain insights into this hypothesis, prescriptions of ABs and 5ARIs were investigated separately. As shown in Figure 3C, up to four ABs were prescribed to the same man. The percentage of men prescribed a single AB was 73.8% and 85.5% (arithmetic mean) when younger (<70 years of age) and older (≥70 years of age) subjects were considered, respectively. Prescription of 5ARIs showed a comparable trend, despite changes that were less pronounced than of ABs (Figure 3D).

These results suggest that prescription of drugs to treat BPH/BPO-associated LUTS could be influenced by factors associated with the age of men, including clinical conditions and life transitions.

## 4. Discussion

BPH/BPO represents a major public health issue because of its prevalence, progressive nature and associated costs [28]. Numerous drugs have proved to exert beneficial effects on the disease, and current guidelines recommend their use for the treatment of BPH/BPO-associated LUTS [3,4]. However, discrepancies exist between the guidelines and actual clinical practice [29]. Drug utilization research could be helpful to better understand prescription behavior, patient preferences, and, primarily, to identify unmet needs, as well as areas of intervention for the healthcare systems.

In Italy, all citizens have access to healthcare services. Pharmaceutical services are provided for free or at a minimum charge as part of the Italian NHS. In addition, data on drug prescriptions reimbursed by the Italian NHS are routinely collected by healthcare authorities at the local, regional and national level. These belong to real-world data that can be analyzed to provide insights into specific questions regarding how drugs are being used. In particular, the use of drugs for the treatment of BPH/BPO-associated LUTS has already been investigated in the Italian population [19,20], and the Italian Medicines Agency(Medicines Utilisation Monitoring Centre) produces and periodically revises reports such as the National Report on medicines use in Italy [30]. Our study of over 4000 men can provide more details on this topic.

Continuous use of medications for the treatment of BPH/BPO-associated LUTS decreases over time [15,16,17,18,19,20,21,22]. Our findings confirm and extend these results. In particular, the overall percentage of men still on treatment at 1 year was 34.5%. However, significant differences were found when different drugs to treat BPH/BPO-associated LUTS were compared. Median time to discontinuation was significantly longer for alfuzosin and dutasteride than for other drugs belonging to the same pharmacological classes. In addition, death was the cause of discontinuation for only 48 men. Thereby, the reasons for non-persistence remain largely elusive. In chronic, long-term, non-life-threatening conditions such as BPH/BPO, the decision to persist with medications mainly depends on patient perceptions of discomfort, symptom control and inconvenience (experience). Thereby, the symptomatic relief perceived by some men after a few weeks or months of treatment may lead them to discontinue the medication because the bothersome condition has disappeared. On the other hand, unmet expectations in terms of symptom control and/or inconvenience may negatively impact the patients’ decision to persist. Patients’ perceptions also include the side effects exerted by the drugs. Tolerability of drugs for BPH/BPO-associated LUTS has been extensively investigated and it has been established that the use of both ABs and 5ARIs is associated with a range of side effects [31]. In particular, ABs could cause cardiovascular events (e.g., postural hypotension, asthenia and dizziness), ejaculatory dysfunction (e.g., retrograde ejaculation) and intraoperative floppy iris syndrome (IFIS). The use of nearly all ABs has been associated with adverse cardiovascular effects, although those related to alfuzosin, tamsulosin and silodosin are relatively weaker than those related to doxazosin and terazosin [32]. Tamsulosin and silodosin also have the most significant association with retrograde ejaculation [33,34]; in contrast, it has been reported that alfuzosin could improve ejaculatory function [35]. IFIS could occur in patients taking ABs who undergo cataract surgery [36]. Therefore, side effects associated with the use of ABs may explain the persistence rates found in the study population. The use of 5ARIs has been associated with decreased libido, gynecomastia and erectile dysfunction. Finasteride exerts fewer sexual side-effects and breast complications than dutasteride when used in the treatment of BPH [37]. As shown in some reports [38], dutasteride could improve the prognosis of patients with prostate cancer and finasteride exerts comparable effects in terms of reducing the incidence of prostate cancer, although it may increase the risk of high-grade Gleason prostate cancer [39]. Since no obvious differences have been reported between dutasteride and finasteride in terms of either efficacy or safety, it seems unlikely that side effects could explain the differences in persistence rates observed in this study.

An association between continuous use of these medications and age-related factors has been suggested by previous studies. However, contrasting results were reported and no definitive conclusion can be drawn. For example, as reported by Nichol et al. [15], younger men were more likely to discontinue treatment with ABs and finasteride. Consistently, Eisen et al. [22] found that older age was associated with better medication adherence and longer persistence to dutasteride and tamsulosin combinations. In contrast, no association between age and persistence to ABs and 5ARIs was reported by Koh et al. [18]. With this study, it was shown that median time to ABs discontinuation was significantly longer for older men than for younger ones. Moreover, a sharp increase (approximately 12.0%) in the proportion of men prescribed a single AB was determined by comparing younger and older men. These unexpected findings imply that long-term continuation of medications to treat BPH/BPO-associated LUTS may depend on factors beyond those aforementioned. Retirement would appear to be an age-related factor possibly associated with the use of ABs. In fact, BPH/BPO is a chronic disease that limits the activities of daily living and degrades the individual’s subjective health status, due to the physical consequences of urinary tract infections and odors. Specifically, it affects health-related QoL by inducing psychological stress in men caused by anxiety, depression and deterioration of their social function [40,41]. Retirement represents a life transition that has been associated with self-rated health status. In addition, it has been associated with changes in the use of antidepressants, antihypertensives, oral antidiabetic drugs and statins [42,43,44,45,46]. Therefore, it is plausible that the use of drugs to treat BPH/BPO-associated LUTS may also be affected by retirement.

### 4.1. Strengths and Limitations

This is the first study investigating the use of drugs to treat BPH/BPO-associated LUTS in a population of the Piedmont Region (Italy). Our results contribute to the framework of existing knowledge on the use of drugs for the treatment of BPH/BPO-associated LUTS. In addition, our findings shed light on some factors influencing persistence to these medications: drug choice and age of men. The major strengths of this study include: (a) the large sample size of over 4000 men; (b) the high quality of real-word data (dispensing records); (c) and the method of analysis. All these elements allowed us to draw robust conclusions.

A major limitation common to retrospective observational studies using administrative data (which are not designed for specific research investigations) was the lack of availability of in-depth clinical information on diagnosis, disease severity and laboratory data (e.g., prostate-specific antigen). Moreover, there was no information available on the proper intake of medications according to the treatment regimen (all prescriptions are administered orally and should generally be taken once a day). Persistence to medications was calculated based on the measured treatment duration, and it was assumed that all drugs dispensed to the subjects analyzed were taken properly. Thus, persistence rates may be overestimated due to this assumption.

The low number of drug classes investigated represents another limitation of this study. In particular, drugs recommended for the treatment of BPH/BPO-associated LUTS belonging to pharmacological classes other than ABs and 5ARIs cannot be investigated for several reasons. For example, a total of 82 men were new users of oxybutynin, and some of them were also prescribed ABs and/or 5ARIs; however, given that, in Italy, reimbursement of muscarinic receptor antagonist prescriptions is limited to patients affected by urgency urinary incontinence related to neurologic diseases, use of oxybutynin cannot be attributed accurately to treatment of BPH/BPO-associated LUTS. In addition, prescription data for some drugs to treat BPH/BPO-associated LUTS (i.e., herbal preparations) are not regularly collected by healthcare authorities, and therefore no solid conclusions can be drawn regarding their use.

### 4.2. Implications for Practice/Further Research

Together with previous results, these findings highlight needs and areas of uncertainty that future studies could address.

Twelve-month persistence rates were 25.5% (95% CI, 24.2 to 26.9%) and 36.0% (95% CI, 33.6 to 38.5%) for AB and 5ARI classes, respectively. Even more pronounced differences were found when single drugs were compared. Consistently, persistence rates in the AB group were lower than those in the 5ARI group [47]. Therefore, these results imply that persistence and/or adherence to medications to treat BPH/BPO-associated LUTS likely depend on a combination of numerous factors, with a wide variation among different drugs and individuals.

Prescription of drugs to treat BPH/BPO-associated LUTS is a complex topic, both due to the number of drugs and the treatment options available. In particular, in the population studied, switches among different drugs belonging to the same pharmacological class occurred commonly. We can argue that aforementioned factors associated with the patient’s perception/preference/experience could explain the decision to switch from one treatment option to another. Moreover, the percentage of men prescribed a single drug and persistence rate to that drug appear to be related measures in this study. Thus, it is plausible that prescription patterns depend on patient experience. Combination therapies were prescribed only to a minor percentage of the study population. This likely reflects physicians’ attitudes or low patient persistence to this treatment option. However, additional factors, including the cost of medications, could not be excluded. For example, in the ASL TO4, prevalent users of silodosin increased over the study period (data not shown). Notably, due to the expiry of the silodosin patent in November 2018, a decrease in the average cost per DDD was measured compared to the previous year, and equivalent (generic) products were launched on the market. Reasonably, these changes may have increased prescriptions of this drug.

Poor persistence to drugs to treat BPH/BPO-associated LUTS has been associated with less favorable clinical outcomes [15,19,20]. As demonstrated by our study, a large proportion of men discontinued drug treatment after the first few fills of their prescription. Therefore, these results highlight unmet needs and areas of intervention for healthcare systems aimed at improving quality of care in the ASL TO4, as well as areas of uncertainty with regarding the long-term effectiveness of these treatments. Additional efforts should be made in order to personalize patients’ therapies by anticipating the growing knowledge on this topic. Men prescribed medications to treat BPH/BPO-associated LUTS should understand the benefits, risks and endpoints of continuous treatment. An effort should be made by physicians (including both general practitioners and urologists) and pharmacists to convey these concepts to patients. Furthermore, the long-term (≥3 years) discontinuation rate and effectiveness of these drugs need to be better evaluated. Interestingly, as reported by de la Rosette et al. [48], the re-initiation rate could be measured from administrative records and used to quantify the success of treatment of BPH/BPO-associated LUTS.

Finally, future research could explore the role of additional factors possibly influencing persistence to these medications, as well as the long-term effects of interventions aimed at improving quality of care in this therapeutic area.

## Figures and Tables

**Figure 1 healthcare-10-02567-f001:**
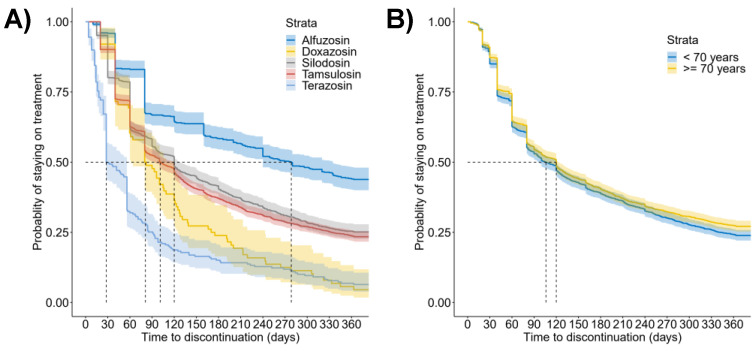
Persistence to treatment with ABs. (**A**) Kaplan-Meier curves of the time to discontinuation for specific drugs (alfuzosin, *N* = 584; doxazosin, *N* = 88; silodosin, *N* = 1074; tamsulosin, *N* = 2057; terazosin, *N* = 219); (**B**) Kaplan-Meier curves of the time to discontinuation for younger (<70 years of age, *N* = 1543) and older (≥70 years of age, *N* = 1730) men.

**Figure 2 healthcare-10-02567-f002:**
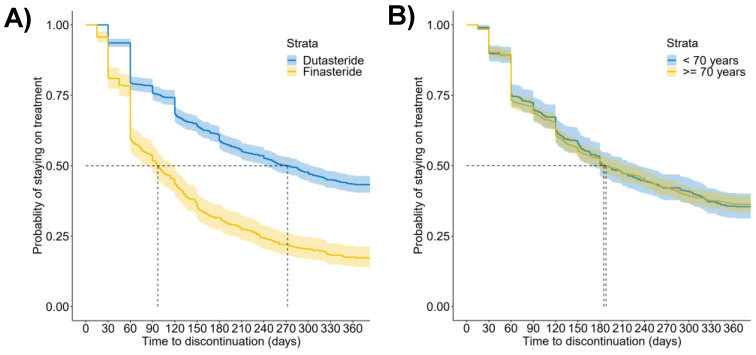
Persistence to treatment with 5ARIs. (**A**) Kaplan-Meier curves of the time to discontinuation for specific drugs (dutasteride, *N* = 1064; finasteride, *N* = 416); (**B**) Kaplan-Meier curves of the time to discontinuation for younger (<70 years of age, *N* = 420) and older (≥70 years of age, *N* = 987) men.

**Figure 3 healthcare-10-02567-f003:**
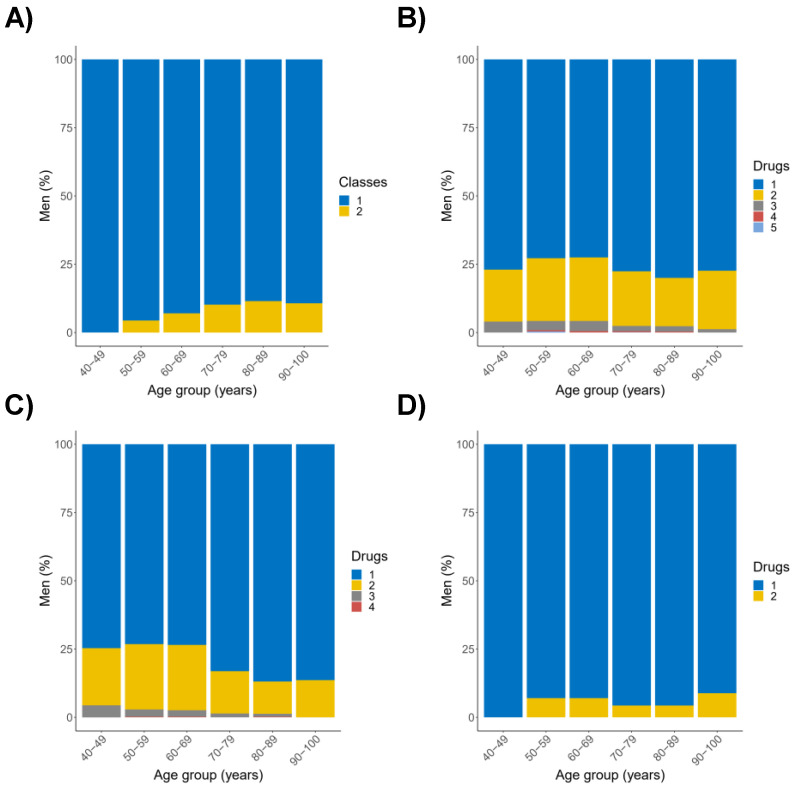
Prescription of drugs for the treatment of BPH/BPO-associated LUTS in the pre-specified age groups. (**A**) Distribution of men with respect to the number of pharmacological classes (Classes) prescribed; (**B**) Distribution of men with respect to the number of drugs (Drugs) prescribed; (**C**) Distribution of men with respect to the number of ABs (Drugs) prescribed; (**D**) Distribution of men with respect to the number of 5ARIs (Drugs) prescribed.

**Table 1 healthcare-10-02567-t001:** General characteristics of the study population.

Group	*N* (%)
Study population	4309 (100.0)
Age (years)	
40–49	100 (2.3)
50–59	525 (12.2)
60–69	1229 (28.5)
70–79	1533 (35.6)
80–89	838 (19.4)
90–100	84 (1.9)

**Table 2 healthcare-10-02567-t002:** Persistence to ABs and to 5ARIs at 365 days.

Category	Percentage of Men Still on Treatment (95% CI)
ABs	25.5 (24.2–26.9)
Alfuzosin	43.8 (39.8–47.9)
Silodosin	25.1 (22.5–27.7)
Tamsulosin	23.4 (21.6–25.3)
Terazosin	6.4 (3.2–9.6)
Doxazosin	4.5 (0.2–8.9)
5ARIs	36.0 (33.6–38.5)
Dutasteride	43.3 (40.3–46.3)
Finasteride	17.3 (13.7–20.9)

CI: confidence interval.

**Table 3 healthcare-10-02567-t003:** Drugs prescribed and age of men.

Drug (*N*)	Median Age [IQR] (Years)	*p*-Value ^a^
Overall (4380)	71.0 [64.0–78.0]	
ABs (3273)	70.0 [63.0–77.0]	NA
Alfuzosin (584)	68.0 [61.0–75.0]	
Doxazosin (88)	73.5 [64.0–78.3]	
Silodosin (1074)	70.0 [63.0–77.0]	
Tamsulosin (2057)	70.0 [63.0–77.0]	
Terazosin (219)	68.0 [59.5–76.0]	
5ARIs (1407)	75.0 [68.0–81.0]	<0.001
Dutasteride (1064)	75.0 [68.0–80.0]	
Finasteride (416)	75.0 [68.0–81.0]	

^a^ For the comparison ABs (control) vs. 5ARIs (Wilcoxon rank sum test). IQR: interquartile range; NA: not applicable.

## Data Availability

Not applicable.

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
