# Peer review of "Persistence to Medications for Benign Prostatic Hyperplasia/Benign Prostatic Obstruction-Associated Lower Urinary Tract Symptoms in the ASL TO4 Regione Piemonte (Italy)"

_healthcare, 2022, doi:10.3390/healthcare10122567_

Round 1

Reviewer 1 Report

The main defect in this manuscript is the lack of sufficient literature reviewing for the previous studies similar to this study, and need more deep discussion for the research results 

Reviewer 2 Report

Comments for the research article titled: Persistence to Medications for Benign Prostatic Hyperplasia/Benign Prostatic Obstruction-Associated Lower Urinary Tract Symptoms in the ASL TO4 Regione Piemonte (Italy)

The authors studied the persistence on the use of medication to treat benign prostatic hyperplasia (BPH)/benign prostatic obstruction (BPO)-associated lower urinary tract symptoms (LUTS) retrospectively in a cohort of 4,309 men. The main focus was on the use of α1-adrenoceptor antagonists (ABs) and steroid 5α-reductase inhibitors (5ARIs) in the ASL 21 TO4, a Local Health Authority in the northern area of the city of Turin (Italy). The study is relevant and provides important information to this topic; however, a few comments:

Overall, the manuscript has several sections duplicated that make it difficult to understand the flow of the article. This includes whole paragraphs and figure 3.

Line 136: There is missing information on the beginning of the paragraph.

Lines 146-154 are duplicated on further down in the manuscript. These also reference Figure 3 which is the last figure in the manuscript.

It would also be useful if the authors include a table with the information regarding the number of patients with one or more prescriptions by age range and their respective percentages. Although he authors describe these percentages in the text, it would be easier for the reader to see it in a table.

Figure 3: The appropriate format to present distribution of subjects should be in a histogram, instead of this type of graph.

Figure 1 and 2: Where these analyses performed with men receiving 1 or more prescriptions, or all of the patients taking the different drugs? Also, it would be helpful if the number of patients were included on the legend of the strata.

It would be important for the authors to state that all the prescriptions are orally administered and taken once a day. This would add consistency to the findings for readers who are not familiarized with these drugs. Also, it would be interesting to know whether the potential side effects could explain the differences observed on the persistence of the different drugs.

Reviewer 3 Report

The authors present an important analysis from real life data in a large sample size. I have the following comments:

1.     Top sentence 136 on page 4 states ‘were prescribed with drugs belonging to a single pharmacological class. At least 1. It seems that some text is missing in this sentence.

2.     Sentence 149 on page 4 states ‘To insight into this hypothesis, prescriptions 149 of ABs and 5ARIs were investigated separately.’ Maybe the authors like to check if the presentation should be changed into ‘To obtain insight into this hypothesis, prescriptions 149 of ABs and 5ARIs were investigated separately.’ or something similar.

3.     Sentence 229-232 states ‘These results suggest that prescription of drugs to treat BPH/BPO-associated LUTS could be influenced by factors associated with the age of men, including clinical condition and life transition.’ I wonder if this statement should rather be positioned in the discussion section than in the result section.

4.     Interestingly a similar analysis was conducted in a single institute. They reviewed the files of 316 patients with lower urinary tract symptoms treated at their department with the alpha-blockers terazosin, alfuzosin or tamsulosin. Using followup data up to 3 years, they calculated re-treatment percentages in each treatment group. Using extended followup of 5 years, they calculated the predictive value of various baseline characteristics for re-treatment. They concluded that patients given alpha-blockers for lower urinary tract symptoms have a high risk of re-treatment. Tamsulosin has a markedly lower re-treatment percentage than alfuzosin and terazosin. Severe symptoms, poor urine flow, an enlarged prostate and urodynamically proven bladder outlet obstruction increases the risk of treatment failure. Preselection of the most suitable candidates for alpha-blockade may reduce this risk (de la Rosette J et al, J Urol 2002 Apr;167(4):1734-9.)

Although the population studied is different as well as the sample size, it is interesting to note that the outcomes are different. How can this be explained? It would be wonderful if the authors could address this in their discussion section. This suggest also to re-direct a little bit the discussion. 

Round 2

Reviewer 1 Report

Author performed the required comment, paper now suitable for publication

Author Response

Thank you for your review

Reviewer 2 Report

The authors have addressed all of my comments and suggestions successfully. I believe that this has improved the manuscript in terms of readability and visual representation of the results obtained. 

Author Response

Thank you for your review